# Quantum Reinforcement Learning for Coordinated Satellite Systems

Gyu Seon Kim[1][0000−0002−5559−9749], Samuel Yen-Chi Chen[2][0000−0003−0114−4826], Soohyun Park[3][0000−0002−6556−9746], and Joongheon Kim[1][0000−0003−2126−768X]

[1] Korea University, Seoul 02841, Korea
{kingdom0545,joongheon}@korea.ac.kr
[2] Wells Fargo, New York, NY 10017, USA
yen-chi.chen@wellsfargo.com
[3] Sookmyung Women's University, Seoul 04310, Korea
soohyun.park@sookmyung.ac.kr

**Abstract.** Reinforcement learning (RL) based on classical neural networks (NN) has demonstrated remarkable advancements across diverse domains. Despite this progress, classical RL encounters training difficulties in systems characterized by high-dimensional action spaces, such as coordinated mobility and satellite systems. In these complex settings, the rapid growth in computational resources required due to increased model parameters substantially limits scalability and convergence speed. Quantum reinforcement learning (QRL), which utilizes quantum neural networks (QNN), offers a promising solution by leveraging quantum mechanical properties, such as superposition and entanglement. QNN particularly enables compact representation of multiple states simultaneously using fewer quantum bits (qubits), drastically reducing computational demands. Owing to its distinct features of rapid convergence and enhanced scalability, QRL emerges as a suitable alternative to classical RL approaches for coordinated mobility and satellite applications. Furthermore, the proposed QRL framework effectively alleviates the curse of dimensionality through efficient utilization of qubits.

**Keywords:** Quantum Reinforcement Learning (QRL) · Quantum Neural Network (QNN) · Mobility · Satellite Systems.

## 1 Introduction

Reinforcement learning (RL) utilizing conventional neural networks (NN) has progressed significantly across various application domains. However, it faces several inherent structural limitations, particularly in handling high-dimensional data and complex decision-making tasks. In high-dimensional environments such as coordinated mobility/satellite systems, higher-dimensional state spaces (inputs of NN) and action spaces (outputs of NN) pose significant challenges to the training performance of conventional RL. In conventional RL, as the dimensions of the state and action spaces increase, the number of parameters that

the model needs to train grows exponentially. This, in turn, leads to a substantial rise in computational costs. Furthermore, data sparsity in high-dimensional spaces necessitates more training samples to develop optimal policies effectively. Consequently, as the action dimension of the agent increases, RL based on conventional artificial NNs suffers from the so-called *curse of dimensionality*, which hampers both training convergence and scalability [1, 2].

Quantum reinforcement learning (QRL) [3] and quantum multi-agent reinforcement learning (QMARL) [4] are emerging as promising approaches to addressing the challenges associated with conventional RL. Developments in quantum computing are opening up innovative possibilities in artificial intelligence (AI), particularly in RL [5, 6]. Quantum AI using quantum neural networks (QNN) leverages fundamental principles of quantum mechanics [7, 8]—such as *superposition* and *entanglement* to overcome the inherent structural limitations of conventional NN [9–11]. Quantum AI can effectively tackle the challenges mentioned above by utilizing these quantum characteristics. QNN can exploit the superposition of *quantum bits (qubits)* to represent multiple possible states at once. This capability allows a single qubit to simultaneously encode multiple states, enabling the efficient representation of high-dimensional data using fewer qubits. Consequently, the resources required to solve high-dimensional problems are greatly minimized, resulting in faster and more efficient training processes. These QNNs have the advantage of allowing QRL and QMARL to be utilized for coordinated mobility/satellite systems. As the number of agents and coordinated mobilities/satellites increases, the agents' action dimensions increase, making it difficult for them to train. However, QRL and QMARL can take advantage of superposition and entanglement phenomena to address this problem through the advantages of *i) fast convergence* and *ii) high scalability*. In particular, the agent's output dimension is extended with only a few qubits by utilizing basis measurement during the measurement phase. This paper introduces the basic concept and structure of QNN and how it can be applied to coordinated mobility/satellite systems in terms of QRL and QMARL. In addition, this paper discusses the areas where QRL and QMARL can be applied.

The main contributions of the proposed QRL framework in this article are as follows. Firstly, this paper utilizes basis measurements to free agents from the curse of dimensionality in high-dimensional environments such as coordinated mobility/satellite systems. It boasts high scalability in response to the agent's high action dimensions with only a few qubits. Secondly, this paper describes the advantages of QRL using QNN with superposition and entanglement, as well as the fundamentals and structures of QNN.

## 2   Related Work

QMARL is suitable for mobility systems as it requires fast convergence, high scalability, and fewer training parameters than conventional RL [4, 12]. Fewer training parameters can exert great power on reusable space rockets, where lightweight and computational simplification are essential, such as Falcon 9 on

Space X [13]. QMARL can be leveraged in rockets and aerial mobility systems such as UAVs, improving training speed and wireless service quality [14, 15]. In smart factory management, QMARL is also used to coordinate Internet-connected multi-robot [16].

## 3    Advantages of Quantum Reinforcement Learning

### 3.1    Fast Convergence

QRL, using QNN, employs the *parameter shift rule (PSR)* for training. QRL using PSR-based QNN have better generalization capabilities [17]. Consequently, QRL training can be executed much more rapidly than conventional RL training [18]. This acceleration is particularly advantageous for real-time scheduling/training within network services and coordinated mobility/satellite systems, where timely updates are critical. Thus, the ability to train each QNN quickly is not just beneficial but essential for effective real-time operations.

### 3.2    High Scalability

QNN can significantly enhance their output dimension, *i.e.*, action dimension of the agent, by incorporating *basis* measurements, thereby overcoming the qubit limitations typical of the noisy intermediate-scale quantum (NISQ) era [19]. In multi-agent reinforcement learning (MARL), the potential number of actions of the agent can significantly increase, necessitating a corresponding rise in the number of qubits required. This increase in action dimension degrades the efficiency of MARL training methods in a finite qubit number environment in the NISQ era. To tackle this challenge, a novel QMARL-based scheduler has been designed using *basis measurements* to achieve a logarithmic reduction in qubit requirements relative to the number of possible actions [20]. This design is crucial for efficiently managing large-scale systems with extensive mobility/satellite bases, minimizing qubit use while maintaining high scalability. Such an approach is particularly beneficial in expansive multi-agent environments with large-scale action dimensions like those involving mobility/satellite, where managing large numbers of agents and action dimensions is critical [21].

## 4    Quantum Neural Networks

### 4.1    Basic Description of Quantum Computing

In QNN, unlike conventional NN, training utilizes units known as *qubits* instead of bits. Qubits, the fundamental units of information in quantum computing, differ from classical bits in that a register of $\mathcal{C}$ classical bits can represent any one of $2^{\mathcal{C}}$ possible states at a time, with each state represented as a vector where only one element is '1' and all others are '0'. Conversely, in quantum mechanics, a quantum state comprising $\mathcal{P}$ qubits is depicted as a complex vector of $2^{\mathcal{P}}$

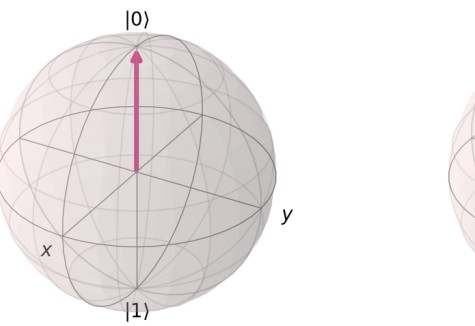

(a) $|0\rangle$ basis in Bloch sphere.          (b) $|1\rangle$ basis in Bloch sphere.

Fig. 1: Quantum states in Bloch sphere.

dimensions. This allows for a quantum state as a superposition of multiple states simultaneously, a phenomenon known as *quantum superposition*. In this paper, qubits are conventionally represented in two fundamental states using the bra-ket notation: $|0\rangle := \begin{bmatrix} 1 \\ 0 \end{bmatrix}, |1\rangle := \begin{bmatrix} 0 \\ 1 \end{bmatrix}$, Moreover, a single qubit state can be expressed as a normalized two-dimensional complex vector: $|\psi\rangle = \mathfrak{E}|0\rangle + \mathfrak{R}|1\rangle = \begin{bmatrix} \alpha \\ \beta \end{bmatrix}$, where $\mathfrak{E}$ and $\mathfrak{R}$ are complex probability amplitudes corresponding to the states $|0\rangle$ and $|1\rangle$, respectively, and must satisfy the normalization condition $|\mathfrak{A}|^2 + |\mathfrak{E}|^2 = 1$. Quantum states are graphically represented within the Bloch sphere in the 3D quantum state space, or Hilbert space, as: $|\psi\rangle = \cos\frac{\theta}{2}|0\rangle + e^{i\phi}\sin\frac{\theta}{2}|1\rangle$, where $\phi$ and $\theta$ are parameters that define the probabilities of measuring states $|0\rangle$ and $|1\rangle$, constrained by $0 \le \theta \le \pi$ and $0 \le \phi < 2\pi$. Here, the basis of the quantum state, $|0\rangle$ and $|1\rangle$, are geometrically represented in the Bloch sphere by Fig. 1(a) and Fig. 1(b), respectively. For a system with $\mathcal{P}$ qubits, quantum states in the Hilbert space are denoted as, $|\psi\rangle = \sum_{\zeta=0}^{2^{\mathcal{P}}-1} \nu_\zeta |\zeta\rangle$, where $\nu_\zeta$ denotes the probability amplitude for each $\zeta$-th basis state, satisfying $\sum_{l=0}^{2^{\mathcal{P}}-1} |\nu_\zeta|^2 = 1$.

### 4.2    Structure of Quantum Neural Networks

As illustrated in Fig. 2, QNN is structured into three distinct phases, *i.e.*, *i) state encoding, ii) parametric quantum circuit (PQC)*, and *iii) measurement* [22].

**State Encoding.** In QRL, the process known as *state encoding* involves translating the states of the environment, typically represented as vectors in conventional RL, into quantum states suitable for quantum computation. In other words, it means encoding existing classical state information into a quantum state. This initial step is crucial for leveraging the potential quantum advantage,

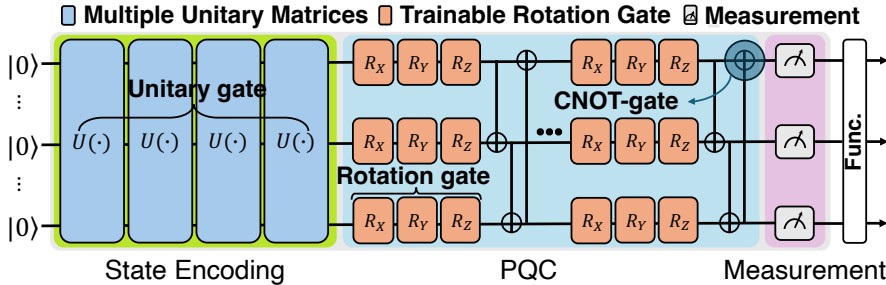

Fig. 2: The structure of QNN

as it significantly influences the quantum system's capacity to depict and manipulate complex environments. Effective state encoding allows quantum computers to process environmental states more quickly and accurately. Additionally, it leverages quantum mechanical advantages like *entanglement* and *superposition* to address more complex problems than conventional RL algorithms can manage. In angle encoding, data is encoded into the angles used in *quantum gate rotations* [23]. This method adjusts the quantum states of qubits using unitary and rotational gates, *e.g.*, $R_X$, $R_Y$, $R_Z$, and is suitable for continuous values, offering a way to represent complex patterns or continuous spaces.

**Parameterized Quantum Circuits.** PQC forms the core structure of QNN in QRL, similar to how neurons and synapses function in conventional NN [7, 24]. PQC comprises quantum gates with adjustable parameters that are fine-tuned during training. In QRL, these circuits transform an encoded quantum state into a new state that represents the policy or value functions relevant to RL tasks [25–27]. The parameters within PQC are analogous to conventional NN weights and optimized using environmental feedback to enhance policy decision-making. PQC incorporates both *rotation gates*, *e.g.*, $R_X$, $R_Y$, $R_Z$, and *entanglement gates*, *e.g.,* controlled-X (CNOT gate), which manipulate the quantum state. The selection and configuration of gates play a crucial role in determining the QRL's training effectiveness [17].

**Measurement.** In QRL, measurement is the process that converts the quantum states manipulated and evolved by PQC back into classical information. In other words, it means decoding an existing quantum state into classical action distribution. This information is then used to determine the *actions* to be executed in the environment. Measurement is essential for translating the outcomes of quantum computations into a form that can be practically utilized for decision-making. When measurement occurs, the quantum state *collapses* into one of the possible basis states, with the specific outcome determined by the probabilities defined by the preceding quantum computations. The result of this measurement is interpreted as an action or a set of actions within the RL.

## 5    Quantum Reinforcement Learning for Coordinated Mobility/Satellite Systems

### 5.1    Parameter Shift Rule for Fast Convergence

The networks considered in coordinated mobility/satellite systems are formulated as multi-agent systems primarily due to their reality. The control tower, *e.g.*, ground station (GS), base station (BS), and leader mobility, corresponds to the $i$-th agent with its own QNN-based RL policy, *i.e.*, $\pi(\mathcal{A}(t)|\mathcal{S}_i(t); \boldsymbol{\theta}_i)$, where $\boldsymbol{\theta}_i$ denotes the parameter of NN. During training, a single centralized critic, with parameters denoted as $\phi$, assesses the value of the policies of multiple actors by approximating the *state-value function*, *i.e.*, $V_\phi(\mathcal{S}(t))$. Here, $\mathcal{S}(t)$ refers to the ground truth state, encompassing all available environmental information [28]. In contrast, each actor independently makes decisions based on its own *partial* observation of the state, indicated as $\mathcal{S}_i(t)$. This training process enables all actors to develop policies for cooperative decision-making, even when each actor can only access *partial* information from the environment. Additionally, during the inference phase, this cooperative approach's distributed nature facilitates effective scalability and efficient use of computing resources. Using the temporal difference (TD) error, multi-agent policy gradient methods are applied to train the quantum multiple-actor centralized-critic networks. The objective function for the $i$-th actor, denoted as $\mathcal{J}(\boldsymbol{\theta}_i)$, can be as,

$$\nabla_{\boldsymbol{\theta}_i} \mathcal{J}(\boldsymbol{\theta}_i) = \mathbb{E}_{\mathcal{S}}\left[\sum_{t=1}^{T}\sum_{i=1}^{N} \delta_{\boldsymbol{\phi}}(t) \cdot \nabla_{\boldsymbol{\theta}_i} \log \pi(\mathcal{A}(t)|\mathcal{S}_i(t); \boldsymbol{\theta}_i)\right], \tag{1}$$

where $\delta_{\boldsymbol{\phi}}(t)$ denotes the TD error. This approach ensures that each actor's policy is optimized based on the observed TD error, thereby enhancing the cooperative multi-agent system's overall performance. The loss function for the critic, denoted as $\mathcal{L}(\phi)$, can be expressed as, $\nabla_\phi \mathcal{L}(\phi) = \sum_{t=1}^{T} \nabla_{\boldsymbol{\phi}} \|\delta_{\boldsymbol{\phi}}(t)\|^2$, where $\delta_{\boldsymbol{\phi}}(t)$ can be expressed as, $\delta_{\boldsymbol{\phi}}(t) = V_\phi(\mathcal{S}(t)) - \hat{V}(t)$, where $V_\phi(\mathcal{S}(t))$ is the estimated state-value function by the critic with parameter $\phi$, and $\hat{V}(t)$ is the target value, typically computed using the TD target. This loss function aims to minimize the difference between the estimated and actual values, thereby refining the critic's ability to evaluate the state accurately. To maximize the objective function for multiple actors and minimize the loss function for the centralized critic, the derivatives concerning the $k$-th parameters of actors and critic are expressed as,

$$\frac{\partial \mathcal{J}(\boldsymbol{\theta}_i)}{\partial \theta_k} = \underbrace{\frac{\partial \mathcal{J}(\boldsymbol{\theta}_i)}{\partial \pi_{\boldsymbol{\theta}_i}} \cdot \frac{\partial \pi_{\boldsymbol{\theta}_i}}{\partial \langle \mathcal{O}_{k,\boldsymbol{\theta}_i}\rangle}}_{\text{(Classical Backpropagation)}} \cdot \underbrace{\frac{\partial \langle \mathcal{O}_{k,\boldsymbol{\theta}_i}\rangle}{\partial \theta_k}}_{\text{(PSR)}}, \tag{2}$$

$$\frac{\partial \mathcal{L}(\boldsymbol{\phi})}{\partial \phi_k} = \underbrace{\frac{\partial \mathcal{L}(\boldsymbol{\phi})}{\partial V_{\boldsymbol{\phi}}} \cdot \frac{\partial V_{\boldsymbol{\phi}}}{\partial \langle \mathcal{O}_{k,\boldsymbol{\phi}}\rangle}}_{\text{(Classical Backpropagation)}} \cdot \underbrace{\frac{\partial \langle \mathcal{O}_{k,\boldsymbol{\phi}}\rangle}{\partial \phi_k}}_{\text{(PSR)}}. \tag{3}$$

In this context, the first and second derivatives on the right-hand side of (2) and (3) can be computed using classical partial derivatives. However, the third derivative cannot be calculated using classical methods because *the quantum state remains unknown until it collapses through measurement*, which is the last stage of the QNN. To address this, the *PSR* is employed for parameter optimization during training [7, 29]. The PSR, when applied to the derivative of the $i$-th actor's $k$-th parameter with respect to the 0-th derivative, is given by, $\frac{\partial \langle \mathcal{O}_{k,\boldsymbol{\theta}_i} \rangle}{\partial \theta_k} = \langle \mathcal{O}_{k,\boldsymbol{\theta}_i + \frac{\pi}{2}\mathbf{e}_k} \rangle - \langle \mathcal{O}_{k,\boldsymbol{\theta}_i - \frac{\pi}{2}\mathbf{e}_k} \rangle$, where $\mathbf{e}_k$ represents the $k$-th basis vector. PSR allows the QNN to be operated under the umbrella of backpropagation or differentiable programming. As a result, this approach allows for faster training in QNN, as described in Sec. 3.

## 5.2 High Scalability for Large-Scale Coordinated Mobility/Satellite Systems

The Pauli-Z measurement evaluates *individual* qubits in quantum states using the Pauli-Z matrix, *i.e.*, $\begin{bmatrix} 1 & 0 \\ 0 & -1 \end{bmatrix}$, where each column corresponds to the computational basis states, specifically $|0\rangle$ and $|1\rangle$. However, in an environment with $\mathcal{P}$ coordinated mobilities/satellites, $2^{\mathcal{P}}$ qubits are still necessary to match the $2^{\mathcal{P}}$ action dimensions required for making combinatorial scheduling decisions for $\mathcal{P}$ coordinated mobilities/satellites. Consequently, the issue known as the 'curse of dimensionality' remains, as this measurement approach does not mitigate the exponential increase in complexity associated with a growing number of coordinated mobilities/satellites [30]. However, with basis measurement, it is possible to compute the probabilities for all $2^{\mathcal{P}}$ combinations using only $\mathcal{P}$ qubits. This is accomplished by measuring the quantum state across all $2^{\mathcal{P}}$ basis, which is expressed as, $\{|\text{Pr}_{\mathcal{B}}(\mathcal{A}_k)\rangle\}_{k=1}^{2^{\mathcal{P}}} \triangleq \left\{ \bigotimes_{k=1}^{\mathcal{P}} |\mho_j^i\rangle \right\}$, where $\mho_j^i$ represents the selection vector of $i$-th control tower for $j$-th mobility/satellite, with $\forall \mho_j^i \in \{0,1\}$ and $\forall j \in [1, \mathcal{P}]$. To summarize, the probability of the $i$-th control tower selecting the $k$-th action based on its policy among $2^{\mathcal{P}}$ possible combinations at time $t$ can be calculated as, $\pi(\mathcal{A}_k(t)|\mathcal{S}_i(t);\boldsymbol{\theta}_i) = \langle\psi|\mathbf{e}_k\rangle\langle\mathbf{e}_k|\psi\rangle = |\langle\psi|\mathbf{e}_k\rangle|^2 = |\alpha_k|^2$, where $|\mathbf{e}_k\rangle\langle\mathbf{e}_k|$ is the projector corresponding to the $k$-th basis, and the set of projectors for all bases is given by $\{|\mathbf{e}_k\rangle\langle\mathbf{e}_k|\}_{k=1}^{2^{\mathcal{P}}}$. Because the probabilities for each action correspond to individual outputs, and the sum of the probabilities of all actions is 1, i.e., $\sum_{k=1}^{2^{\mathcal{P}}} \pi(\mathcal{A}_k(t)|\mathcal{S}_i(t);\boldsymbol{\theta}_i) = 1$.

## 6 Performance Evaluation

The experimental environment has a vast $2^{16}$ action dimension of agents, with 16 mobilities/satellites that agents must coordinate and 4 control towers. In addition, the following hyper-parameters are used in the experiment, *i.e.*, number of qubits (16), training epochs ($10k$), actor and critic's learning rate ($5 \times 10^{-3}$, $2.5 \times 10^{-4}$), initial/minimum/decay rate of exploration ($0.4$, $10^{-2}$, $5 \times 10^{-5}$),

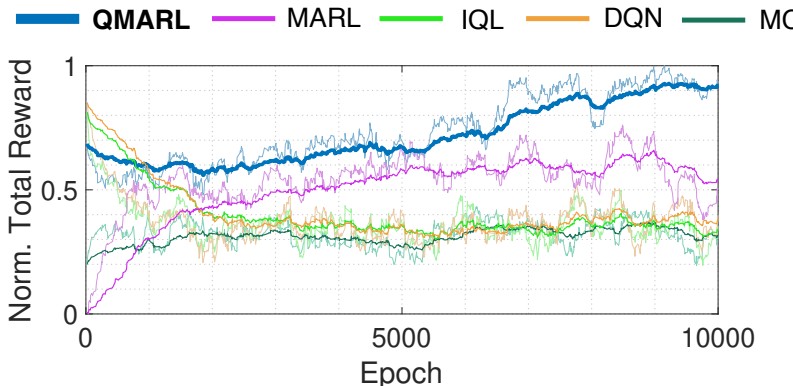

Fig. 3: Normalized reward performance in a coordinated satellite system.

batch size (32), discount factor (0.98), activation function (ReLU), and optimizer (Adam). Agents' actions are to choose which mobility/satellites to receive communication services, and the reward function is designed to maximize the QoS, capacity, and remaining energy of mobilities/satellites. The considered benchmarks are, *i)* MARL (conventional MARL), *ii)* Independent Q-Learning (IQL), *iii)* Deep Q-Learning (DQN), and *iv)* Monte Carlo (MC). Fig. 3 shows the normalized reward for each algorithm. Even in environments with vast action dimensions, such as $2^{16}$, only the QMARL-based scheduler is free from the curse of dimensionality with the highest reward.

## 7   Concluding Remarks

This paper demonstrates that QRL addresses the challenges of conventional RL in environments with large action dimensions, such as coordinated satellite systems. QRL's unique advantages, including fast convergence and high scalability, highlight its potential for effective deployment in complex system operations. In future work, the applications of QMARL for various mobility systems can be considerable.

## 8   Acknowledgments

The full version of this paper was presented at the IEEE International Conference on Acoustics, Speech, and Signal Processing (ICASSP) 2025, held in Hyderabad, India. This work was supported by Institute of Information & Communications Technology Planning & Evaluation (IITP) grant funded by the Korea government [MSIT (Ministry of Science and ICT (Information and Communications Technology))] (RS-2024-00439803, SW Star Lab) for Quantum AI Empowered Second-Life Platform Technology. The corresponding author of this paper is Joongheon Kim (e-mail: joongheon@korea.ac.kr).

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
