# OpenReview forum: "Quantum Reinforcement Learning for Coordinated Satellite Systems"
_purdue.edu/Purdue_University/PQAI/2025/Symposium — PQAI 2025 Oral_

### Official Review · Reviewer_h2Ha · 2025-07-19

**Rating:** 3
**Confidence:** 4

**Review:**

The manuscript deals with an interesting topic and is well written.
An application with 2^16 actions is exciting. At first glance, I like the manuscript very much.

Unfortunately, it is incomplete overall:
* The explanation of the benchmark environment is incomplete and there is also no reference mentioned. In this form, the results can neither be reproduced nor evaluated.
* The description of the algorithm used is incomplete, important details are missing. In this form, the results can neither be reproduced nor evaluated.

Further comments:

In Fig. 3 it is not explained what the thick and thin lines mean for each color.

Recommendation:

Reject

---

### Official Review · Reviewer_vzmT · 2025-07-23
**A well-motivated application of QRL to satellite coordination with promising results, though limited by lack of hardware validation — weak accept.**

**Rating:** 8
**Confidence:** 5

**Review:**

This paper looks at using quantum reinforcement learning (QRL) to handle complex satellite coordination tasks with large action spaces. It uses quantum neural networks (QNNs) to boost scalability and training speed through (1) superposition, (2) basis measurements, and (3) parameter-shift optimization.

Strengths

Well-motivated use of QRL for high-dimensional control.

Clear QNN structure and role in RL.

QRL shows better results than classical baselines.



Weaknesses

No real-device testing or noise discussion.

---

### Official Review · Reviewer_NdQU · 2025-07-25
**Reviewer's feedback**

**Rating:** 6
**Confidence:** 4

**Review:**

This paper explores the application of Quantum Multi-Agent Reinforcement Learning (QMARL) to high-dimensional problems, such as coordinated mobility in satellite systems. The manuscript is well-written and provides a clear and accessible introduction to quantum computing and quantum neural networks, making it suitable for a broad audience. It then proceeds to describe data encoding strategies and the architecture of the quantum model used in the numerical experiments. The authors demonstrate that the QMARL framework achieves superior performance in specific instances of the mobility/satellite coordination problem, particularly in scenarios with $2^16$ dimensionality.

While the results are promising, they are limited in scope. The study focuses on a single dataset with relatively modest dimensionality, especially by quantum standards, raising questions about the generalizability of the findings. Moreover, it is well known that VQAs, particularly those employing hardware-efficient ansätze like the PQC shown in Fig. 2, are prone to barren plateaus. These plateaus can severely hinder training as the number of layers increases, even logarithmically with the number of qubits.

Given these limitations, caution is warranted when applying quantum-classical hybrid models to classical problems such as the one considered in this paper. Demonstrating superior performance in truly high-dimensional regimes would be essential to justify the use of quantum reinforcement learning in practice. Unfortunately, the current study does not address this, leaving open the question of whether QMARL can scale effectively and meaningfully outperform classical approaches.

That said, the paper is thoughtfully composed and contributes to the growing body of literature exploring quantum approaches to complex learning tasks. It may serve as a valuable reference for future work in this area.

---

### Decision · Program_Chairs · 2025-07-29

Accept (Oral)